# Creating a Healthy Environment for Children: GIS Tools for Improving the Quality of the Social Welfare Management System

**DOI:** 10.3390/ijerph19127128

**Published:** 2022-06-10

**Authors:** Alina Źróbek-Różańska, Marek Ogryzek, Anna Źróbek-Sokolnik

**Affiliations:** 1Department of Socio-Economic Geography, Faculty of Geoengineering, Institute of Spatial Management and Geography, University of Warmia and Mazury in Olsztyn, 10-720 Olsztyn, Poland; a.zrobeksokolnik@uwm.edu.pl; 2Department of Land Management and Geographic Information Systems, Faculty of Geoengineering, Institute of Spatial Management and Geography, University of Warmia and Mazury in Olsztyn, 10-720 Olsztyn, Poland; marek.ogryzek@uwm.edu.pl

**Keywords:** children’s health, poverty, public expenditure, social assistance, Geographic Information System

## Abstract

Childhood is considered to be the most vital period for mental, physical, and social development. Even short-term deprivation of nutrition, health care, education, and affection in childhood can have long-term and irreversible negative consequences. Various social assistance programs are being launched around the world to eliminate or alleviate social problems, including those experienced by children in their immediate environment. Different solutions have been proposed around the world, but welfare systems in all countries share the following common features: social assistance is necessary and underfinanced, and social workers struggle to cope with caseloads. As a result, welfare work is stressful and not highly effective. In this study, modern Geographic Information System (GIS) tools for supporting the employees of social assistance centers (SACs) have been proposed. The data relating to welfare beneficiaries were analyzed by nonparametric kernel density estimation and divided into five datasets. The kernel density tool in ArcGIS Pro software (Esri Polska sp. z o.o., Warsaw, Poland) was used to visualize areas with a relatively high prevalence of social problems, as well as areas where the neighborhood can deliver synergistic effects. A multicriteria analysis (MCA) procedure for mapping social problems was proposed, and an algorithm was developed in the GIS environment. The generated maps deliver helpful information for supporting SAC employees, as well as monitoring, planning, and initiating preventive measures. Above all, the presented method was designed to improve living conditions by facilitating the management of welfare workers’ duties. Therefore, the proposed approach had to be effective and easy to use without an advanced knowledge of GIS tools.

## 1. Introduction

Temporary or permanent social problems that are too complex to be individually resolved affect many people in all countries around the world. Poverty is a multidimensional problem that thwarts the social and economic development of many nations and obstructs sustainable development. Individual crises can result from poor life choices but also from circumstances that are beyond individual control [1,2]. Natural aging processes gradually deprive older citizens of their physical and intellectual abilities [3,4]. Rapid social and economic changes can lead to a rapid deterioration in living standards. Many unqualified workers slip into poverty when large production plants that monopolize the local market are closed down [5]. Livelihoods can also be jeopardized by random events such as a sudden illness or an accident. Natural disasters lead to the loss of property or can prevent certain types of economic activity to be carried out in the affected area. Global pandemics and the associated restrictions contribute to economic decline and the collapse of healthcare systems [6,7,8,9]. Some crisis situations are temporary: laid off workers find new jobs, and the damage wrought by natural disasters is repaired. Other events have long-lasting or permanent consequences. Diseases, including alcoholism, can lead to permanent health impairment [10]. Some problems are hereditary and emerge in successive generations. Children exposed to adverse social behaviors are more likely to repeat the same patterns of behavior in adulthood [11]. People affected by permanent or inherited problems should receive help as part of various social assistance schemes.

Various approaches to social assistance and different social support programs exist around the world. Regardless of the guiding principles and the implemented solutions adopted by different countries, social assistance has the following common points: it is essential—welfare programs preserve human dignity and prevent social problems from escalating; it is underfinanced—welfare programs drain the budget without generating quick and satisfactory results. In addition to cash benefits, welfare programs require personal involvement of social workers employed in social assistance centers (SACs). Social assistance programs are long-term schemes which aim to improve the beneficiaries’ skills and qualifications, eliminate harmful behaviors and overcome devastating addictions. These measures require sufficient numbers of qualified social workers and effective organization of their duties. Social assistance institutions cannot reach large numbers of geographically dispersed recipients when they are understaffed. Therefore, social welfare schemes focus on those who most need social protection and on special interventions, whereas preventive measures, although equally important, are often not taken. As a result, SAC employees are under considerable stress and suffer from case overload. Social assistance is a difficult and costly process, which is why modern technological solutions should be introduced to improve the quality of welfare services and lighten the burden of social workers.

The following research hypothesis was formulated to address the above concerns: SACs require technical support, and the quality of social services can be improved by providing SACs with Geographic Information System (GIS) tools for mapping and visualizing the geographic distribution of potential welfare recipients. The main aim of this study was to propose an original GIS-based method for supporting SAC operations. A GIS-based multicriteria decision analysis was performed to map the geographic distribution and concentration of crisis events that should be addressed by social workers. The analysis relied on five sets of criteria to identify specific social problems. Five groups of negative social phenomena were identified based on a review of the literature and empirical research: (1) poverty, (2) risk of poverty, (3) personal assistance from social workers, (4) aging, and (5) domestic violence. The applicability of the proposed method was tested on the example of an SAC in a municipality, which is the smallest unit of administrative division in Poland. Data for the study were obtained from publicly available statistical datasets, the respective departments of municipal offices, SACs, and police stations. The names of the investigated towns and villages were coded to prevent stigmatization. The proposed method is universal and can be applied in all types of territorial units. The presented GIS tools are flexible and can be easily adapted to the needs of other countries by incorporating country-specific welfare data in the analysis. Data can be shared and analyzed using the Infrastructure for Spatial Information in the European Community INSPIRE [12]. In addition to quantitative attributes, such databases should also contain comprehensive spatial information because in the era of globalization, the creation of databases should be standardized [13].

A multicriteria analysis (MCA), also referred to as multiple-criteria decision-making (MCDM), multiple-criteria decision analysis (MCDA), multi-objective decision analysis (MODA), multiple-attribute decision-making (MADM), or multi-dimensional decision-making (MDDM), includes various classes of methods, techniques, and tools of varying complexity that explicitly take into account many goals and criteria (or attributes) in decision-making problems [14]. Data visualization and MCA [15] are integrated in Geographic Information Systems. GIS tools are applied in spatial analyses to generate maps that illustrate the suitability or limitations of specific attributes (criteria) of the examined phenomena. Accessibility analyses provide a convincing theoretical framework for assessing different aspects of the quality of life [16]. The final map describing a given phenomenon is developed by superimposing suitability or constraint maps. The result is a binary (0, 1) or a weighted overlay, with cell values arranged on a single scale. GIS uses the Fuzzy Overlay or Raster Calculator tool to overlay maps. Features and weights are selected by the analyst, and the results are dependent on the applied settings. However, the parameters of individual features do not have to be set when the weights are equal or when binary suitability parameters are used. This approach does not involve a prescribed method for compiling suitability or constraint maps because the reclassify tool supports data transformation on the same scale. Therefore, raw data (such as quantitative data) can be analyzed with the use of spatial analysis tools. However, GIS tools have never been applied to create visualizations supporting the management of welfare workers’ duties. The described method is not new, but it can be adapted to the needs of users who do not have advanced knowledge of GIS, thus contributing to an improvement in the working conditions of social welfare employees. In turn, effective social services will lead to an improvement in the living conditions of people in various crisis situations, thereby creating a healthier and safer environment for children.

Social welfare was an important rationale for undertaking the present study because in addition to explaining and describing specific phenomena, the proposed approach should also serve the community. Therefore, the developed method was designed to be practicable. However, the proposed methodology has to meet several conditions in order to be applicable in practice. The discussed approach for visualizing the severity of social problems is designed for social welfare employees who usually have little or no experience in working with advanced data processing tools. Therefore, the developed tool should be effective, useful, as well as relatively simple and easy to use. The employees of social welfare centers, as well as the supervising institutions (the Supreme Audit Office in Poland), rely on selected indicators to calculate the percentage share of various types of welfare payments and to determine changes in the number of welfare recipients in the analyzed period. These institutions also calculate other indices, including the poverty index and indices describing local levels of social deprivation, structure of welfare recipients, social work, social contracts, needs fulfilment, changes in foster care, and active aging.

## 2. Social Assistance in Various Countries of the World, with Special Emphasis on Poland

### 2.1. Global Perspective

Different cultures and communities have developed various ways of empathizing with and providing social assistance for people in difficult circumstances. In some countries, welfare services are addressed only to individuals who are unable to work due to physical disability. For example, in the Republic of South Africa, social assistance consists mostly of retirement pensions, disability benefits, and child support benefits. Similar to Latin American countries, the South African welfare system relies on the fundamental assumption that social services should be addressed only to physically disabled people, whereas healthy individuals should be able to secure their livelihoods through work [17] and should not be entitled to any financial assistance from the state [18]. A similar approach has been adopted in China, where families are responsible for the well-being of all family members, and where active social support for children and the elderly is deemed to undermine the caregiving role of the family [19]. Despite the above, the Chinese Ministry of Civil Affairs has introduced a minimum living guarantee system for urban residents, which implies that urban households with children, disabled family members, and members who require special care can apply for benefits. The implemented social assistance programs targeted not only disabled citizens, but also individuals experiencing financial difficulties. Under the Constitution of the Federal Republic of Brazil in 1988, social welfare is a duty of the state, and all citizens in need are entitled to social assistance. To fulfill these obligations, the federal government implements various programs that cater to the basic needs of people living in extreme poverty [20]. The “equal opportunities” concept is an alternative approach to social assistance, where everyone has access to social services. In Sweden, all citizens are entitled to social assistance regardless of income. Childcare is heavily subsidized, and healthcare and education are free. Low-income households are also entitled to financial assistance that guarantees a minimum standard of living [21,22,23]. A similar approach has been adopted in Poland where childcare (for children older than three years) is subsidized by the state, primary and secondary education is free, and healthcare services are universally available to all citizens who pay the health insurance premium. In turn, the goal of social welfare is to cater to the basic needs of all citizens and provide them with a dignified standard of living. Social assistance programs also deliver preventive interventions, help vulnerable individuals and families to live independently, and promote social integration of disadvantaged people [24].

Social assistance is provided by SAC employees. Social work is a profession that requires high qualifications, and it is physically and psychologically demanding. Social workers must be sensitive, empathetic, able to effectively plan their operations and work under stress because SACs tend to be understaffed. The British Local Government Association has developed eight standards for supporting the employment of social workers across different agencies, including transparent systems for workload planning and agenda management. Social assistance is also chronically underfinanced. An audit conducted by the Polish Supreme Audit Office in 2018 revealed numerous problems that undermine social workers’ effectiveness. Above all, SACs were understaffed, and the number of cases handled by social workers exceeded their contractual workload three-fold. Welfare needs were not regularly monitored and addressed in real time to prevent future problems. Predictions regarding the number of individuals in need of support were developed based on statistical data within a strict financial framework rather than based on assessments of real-world needs. Information about local needs was not presented in reports or planning documents, which prevented SACs from planning realistic budgets and addressing current needs. Social workers in other countries, including the United Kingdom, face similar difficulties [25]. The implementation of modern tools for diagnosing and anticipating social needs in different areas could significantly improve effective delivery and management of social assistance services.

### 2.2. National Perspective—A Case Study of Poland

In Poland, municipal authorities are responsible for fulfilling local communities’ basic social needs, and this goal is accomplished by dedicated organizations, namely social assistance centers (SACs). Each welfare district is divided into smaller areas where social assistance is provided by dedicated SAC employees. Social workers perform office duties and help welfare recipients directly in their place of residence (within their respective districts). They regularly monitor the beneficiaries’ life situation by visiting homes, performing comprehensive family assessments, working with various institutions, undertaking preventive measures to minimize the risk of social problems, and responding to domestic violence during interventions.

Social assistance consists of various types of services. Individuals and households with incomes below the poverty threshold are eligible to social assistance, including cash benefits, noncash benefits, counseling, and support. Pursuant to the provisions of the Social Assistance Act of 2004, social assistance services are provided to individuals and families affected by poverty, orphanhood, homelessness, unemployment, disability, and violence, to protect human trafficking victims, support large families and destitute mothers, promote social inclusion of young offenders leaving detention facilities, prevent alcoholism and drug abuse, support individuals and families affected by a personal crisis, or a natural or environmental disaster. Individuals and families who are experiencing a crisis for at least one of the above reasons and whose incomes are below the poverty threshold are entitled to cash benefits and social assistance.

Social assistance includes the following:(1)cash benefits (addressed only to individuals whose incomes are below the poverty threshold), including
regular benefits for individuals who are permanently unable to work due to old age or disability;temporary benefits for individuals affected by prolonged illness, disability or unemployment;targeted benefits that meet specific life needs, such as total or partial reimbursement of the costs associated with medication or food purchases;
(2)noncash benefits (which are also available only to individuals whose incomes are below the poverty threshold), including shelter, meals, clothing, personal care, accommodation in welfare homes, and housing benefits (help towards rent).

Social assistance is also provided to individuals and families who are affected by specific problems and are at risk of social exclusion, regardless of income. Beneficiaries are expected to cooperate with social workers to resolve their life difficulties. Benefits and social assistance may be revoked or limited, and cash benefits can be replaced with noncash benefits, when beneficiaries abuse the welfare program. Social workers monitor the recipients’ progress to ensure that the awarded benefits are spent properly. Modern GIS tools can be applied to support SAC employees who regularly visit the beneficiaries’ homes in the process of planning their trips.

### 2.3. Children’s Welfare—A Huge Challenge for Social Assistance

Children are a special category of welfare recipients. As childhood is the most vital period during mental, physical, and social development, short-term deprivation of nutrition, health care, education, and affection in childhood could have long-term irreversible consequences [26]. It is undeniable that poor children are more likely to grow up to become poor adults [27,28]. Many studies over the years have shown that the socioeconomic position of the family, measured by parental education, income, or housing, influences a child’s health and well-being [29,30]. Low family income has a negative impact on cognitive development, and it increases the probabilities of low educational attainment, economic inactivity, and self-support problems in young adulthood. Children need positive role models to adopt constructive social roles and become productive members of society. However, dysfunctional families rarely set positive examples for their children [31]. Therefore, special attention should be paid to a child’s environment. The negative behaviors and patterns observed in dysfunctional families, such as chronic unemployment, alcoholism, or violence, are reinforced in children and tend to be repeated from generation to generation. Moreover, the problem of poverty is chronic and is more likely to replicate as it is inheritable [11] because of family patterns and ‘bad neighborhood effect’ (i.e., absence of good level employment opportunities).

In Poland, family assistants work with dysfunctional families to prevent negative patterns from being repeated by household members, including children. Assistants provide dysfunctional families with support, give advice, and promote positive social behaviors. The main role of a family assistant is to manage and counteract social problems in dysfunctional families. Assistants assume the role of mediators, representatives, and counselors who give support to families and encourage them to actively participate in the process of overcoming their problems. They undertake various measures to change the family dynamics, motivate family members to initiate positive changes, and build a sense of self-worth in family members. Assistants also prepare families to assume age-appropriate social roles; they motivate family members to improve their qualifications and find work. Assistants must be highly motivated to encourage all members of dysfunctional families to take active steps towards improving their fate and rebuilding family ties [31].

Social assistance centers protect disadvantaged children and prevent social problems from propagating. These tasks pose an immense challenge for SAC employees who are not only expected to manage crisis situations, but also to prevent them. To effectively counteract social problems, social workers have to regularly monitor the environment in which children are raised, including their homes (support in the resolution of family problems) as well as the neighborhood (support for entire districts or communities). Maps generated with the use of the proposed method can be very helpful in this process.

## 3. Materials and Methods

The following research hypothesis was formulated: maps presenting the location and intensity of social problems or the risk of social problems in a given area constitute effective tools that support the operations of SACs. Social problems are a complex and multi-faceted issue, which is why numerous, unrelated sources of data had to be obtained for research purposes. As a result, the study was conducted in four stages: I, Data acquisition; II, Development of an integrated database; III, Development of thematically linked datasets; IV, Generation of thematically linked maps in GIS. The research process was designed with the use of the MCA algorithm which is presented in Figure 1.

In stage I, data relating to poverty and other crisis situations were aggregated. In most countries, including Poland, eligibility for social assistance is determined mainly based on income. The Social Assistance Act stipulates that the main goal of social assistance is to support individuals and families affected by poverty, orphanhood, homelessness, unemployment, disability, prolonged or serious illness, domestic violence; to protect human trafficking victims, support large families and destitute mothers, individuals who require assistance with childcare and household matters, in particular in broken, single-parent or large families; to promote the integration of foreigners who have been granted refugee status, subsidiary protection, or a temporary residence permit; to promote social inclusion of offenders leaving detention facilities; to prevent alcoholism and drug abuse, support individuals and families affected by a personal crisis, or a natural or environmental disaster. The existing types and forms of social assistance are prescribed by legal regulations. The provisions of Polish legal acts were analyzed to determine the number of welfare recipients in various categories. The relevant databases are kept by SACs in Polish municipalities. Population data, the number of large families (with at least three children), and the age structure of the population were determined based on the data obtained from the civil registry department of the municipal office. The rate of population aging and the aging index were calculated. Information about the availability of social housing, the number of evictions from social housing, and the number of people who are in arrears with rent payments was obtained from the real estate department of the municipal office. Information about the number of domestic violence interventions and the number of families covered by the Blue Card procedure (families that are registered and monitored due to a history of domestic violence) was acquired from the local police station.

In stage II, the data acquired from the above sources were used to develop an integrated database. The gathered data were divided into thematic groups and linked with welfare districts according to the territorial division system applied by SACs. Five groups of negative social phenomena were identified based on a review of the literature and empirical research.

In stage III, each of the identified crisis phenomena was linked with information about the types of provided social assistance or the description of a social problem. The resulting datasets and their characteristics are presented in Table 1.

Dataset 1 contains information about actual poverty. Poverty was visualized by the number of households receiving income support and housing benefits. Dataset 2 describes the risk of poverty. Each of the described social problems contributes to the risk of a crisis situation. The accumulation of social problems also jeopardizes the well-being of children residing in high-risk areas. For instance, children who grow up in areas affected by alcoholism or drug abuse are more likely to adopt dysfunctional behaviors and carry them into adulthood. Dataset 3 was identified to describe cases where individuals required assistance in coping with childcare or household tasks. Social workers regularly visit such households to provide beneficiaries with help in daily matters. Dataset 4 has prognostic value. Persons living in areas with a high age dependency ratio are likely to require social assistance in the near future. Dataset 5 contains information about domestic violence. Domestic violence is a serious and complex problem which requires preventive measures and, if such measures are ineffective, interventions conducted by social workers with the assistance of the police and healthcare professionals (doctors, paramedics, nurses).

In stage IV of the research procedure, the results of the analyses conducted in the previous stages were visualized on thematic maps. An algorithm for mapping social data was developed by the authors. Kernel density, raster reclassify, and raster calculator tools were used for this purpose. The authors relied on GIS tools that are widely applied in various scientific disciplines to process and analyze spatial data [32]. Based on a review of the literature, social problems were mapped with the use of the multicriteria analysis (MCA) procedure. The algorithm developed in the GIS environment for mapping social problems is presented in Figure 2.

GIS tools are highly useful for visualizing research results on maps [33], and they provide various methods and techniques for data analysis, including multicriteria analysis [34]. In the MCA approach, a finite number of possibilities is analyzed in view of various criteria and objectives [35]. In each group of the factors, the spatial distribution of social problems was determined by nonparametric estimation with the use of the Epanechnikov kernel [36]:(1)$$f_{\lambda }\left(x\right)=\frac{1}{n\lambda }\Sigma k_{0}\left(\frac{x-x_{i}}{\lambda }\right)$$
where*k*_0_–quadratic kernel function*λ*–smoothing parameter


(2)$$k_{0}\left(t\right)=\left\{\begin{array}{c} 0.75\left(1-t^{2}\right)\\ 0\; \end{array}\right.$$
for *t* ≤ 1 in the remaining cases

This function is implemented in the Kernel Density tool in Esri ArcGIS software (Esri Polska sp. z o.o., Warsaw, Poland). The category of point features is selected, and the Kernel Density tool calculates surfaces that are fitted to each point. The Kernel Density tool calculates the density of features in the neighborhood around those features. It can be calculated for both point and line features [37]. Kernel Density calculates the density of point features around each output raster cell. Conceptually, a smoothly curved surface is fitted over each point. The surface value is highest at the location of the point and diminishes with increasing distance from the point, reaching zero at the search radius distance from the point. Only a circular neighborhood is possible. The volume under the surface equals the population field value for the point, or 1 if NONE is specified. The density at each output raster cell is calculated by adding the values of all the kernel surfaces where they overlay the raster cell center. The kernel function is based on the quartic kernel function described by Silverman [38]:(3)$$Density=\frac{1}{{\left(radius\right)}^{2}}\overset{n}{\underset{i=1}{\sum }}\frac{3}{\pi }pop_{i}{\left(1-{\left(\frac{dist_{i}}{radius}\right)}^{2}\right)}^{2}$$
where
*i* = 1,…, *n* are the input points. Only points located within the search radius from location (x, y) are included in the sum;*pop_i_* is the population field value of point *i*, which is an optional parameter;*dist_i_* is the distance between point *i* and the location (x, y).

Kernel Interpolation is a variant of first-order Local Polynomial Interpolation, where a method similar to ridge regression in the estimation of regression coefficients is applied to prevent the instability of the calculations. When the estimate has only a small bias and is much more precise than an unbiased estimator, it may well be the preferred estimator [39]. The Epanechnikov kernel usually produces better results when first-order polynomials are used. However, depending on the data, cross-validation and validation diagnostics may suggest another kernel [40]. Kernel interpolation accounts for natural barriers such as mountains, rivers, and lakes, which is an important functionality. This method calculates the shortest distance between these points. As a result, the points on all sides of nontransparent (absolute) barriers are connected by a series of straight lines [41]. It was assumed that the algorithm for mapping selected social problems would rely on raw quantitative data describing a single feature (Feature). In Table 1, social problems are presented in column 2, and the features that influence these problems are presented in column 3. These features were analyzed with the use of the Kernel Density method and the MCA approach. Individual features were transformed onto the same scale with the use of identical data intervals. The generated data were then overlapped to produce a map illustrating the distribution of social problems.

In the procedure, the aggregated statistical data were used to build a database. The purpose of using specific databases (minimum databases and problem-specific databases) is to ensure systematic and consistent data collection [42]. The reliability of the presented data is determined by their source, whereas the extent to which these data describe the analyzed phenomena is not correlated with their reliability or perceptions of their reliability (Li et al., 2018). Data are georeferenced, and they can be presented on a map with the use of unique points, where the extent to which a given factor contributes to the analyzed phenomena is represented by a color scale. However, this visualization technique does not depict the prevalence of the studied problem, and it does not support analyses of the combined impact of several criteria in a given area. In line with the adopted procedure, individual criteria associated with the analyzed problem (points) were transformed to raster layers in a spatial analysis. The Kernel Density tool in the Spatial Analyst toolbox was used for this purpose. Kernel Density calculates the density of points surrounding each output raster cell. The calculated density is multiplied by the number of points or the sum of the population field. As a result of this correction, the spatial integral is equal to the number of points (or the sum or the population field), and it does not always equal 1. A Quartic kernel is used in this implementation. The formula must be calculated for every location because a raster is created from the calculations applied to the center of each pixel [43]. Each criterion was then reclassified with 3D Analyst tools. The Reclassify tool was selected from the Raster Reclass toolbox. The input raster contains an attribute table, and it was used to create the initial reclassification table on a scale of 1 to 5. In the GIS approach, Raster Reclass facilitates a comparison of numerous maps [44]. Map Algebra is used in the last step of the MCA process. Data were presented on the same scale, and classes of criteria were added to the analyzed area with the Raster Calculator tool. Algebraic expressions can be applied not only to model layers, but also to describe neighborhood and zone variables [45].

Kernel Density maps were developed for each studied feature. Each map was reclassified on the same scale, and in the next step, the maps illustrating the distribution of features assigned to a given social problem (Table 1) were applied to each of the five analyzed phenomena. Mathematical operator symbols must be selected when the algorithm is applied to search for data. The results were divided into five classes. Class 1 denotes areas with the lowest risk of social problems. Class 2 represents areas with a very low risk of social problems. Class 3 denotes areas with a moderate risk of negative phenomena (transitional areas). Class 4 contains areas with an increased risk of social problems. Class 5 represents areas which are at highest risk of social problems. The adopted scale for identifying the severity of social problems supports rational planning of welfare spending and staffing policies in SACs. In the generated maps, the severity of social problems was marked with a blue color gradient. The darker the color, the greater the need for social assistance in a given area.

## 4. Results

The proposed method was used to generate maps presenting five sets of data (1 to 5) in the analyzed area. The spatial distribution of poverty areas is visualized in Map 1 (Figure 3).

Figure 3 was developed in the GIS environment using Map Algebra with the Raster Calculator tool based on the distribution of features 1, 2, 3, 4, 5, 6, 7, 8, 9, 10, 11, 12, 13, 14, 15, 16, 17, 18, 19, 20, 21, and 22 (Table 1) determined with the use of the Kernel Density method. The resulting map visualizes poverty areas (SP 1, Table 1). An analysis of Map 1 indicates that the highest number of individuals entitled to income support reside in welfare districts covering towns and villages with code numbers 4, 16, and 17, as well as 3, 5, 12, 14, 19, 22, and 23. The resulting information can be useful in managing social assistance services, monitoring the operations of social workers and providing them with support in the process of planning trips to areas inhabited by recipients of income support payments. The future risk of poverty is presented in Map 2 (Figure 4).

Figure 4 was developed in the GIS environment using Map Algebra with the Raster Calculator tool based on the distribution of features 12, 13, 14, 15, 16, 17, 18, 19, 20, 21, and 22 (Table 1) determined by the Kernel Density method. The resulting map visualizes the risk of poverty (SP 2, Table 1).

The second dataset was visualized in Map 2 to present areas that may be affected by poverty in the future due to the spatial concentration of risk factors for social problems. It should be noted that Maps 1 and 2 contain the same areas marked in dark blue (4, 16, and 17), but differences between the two maps can also be observed (areas No. 3, 6, and 13). In both maps, areas marked with the darkest shade of blue represent welfare districts affected by permanent poverty. Despite income support payments (which are allocated based on the incomes generated in the previous calendar year), social problems that increase the risk of poverty are still prevalent in these areas. Therefore, these areas should be closely monitored to identify possible solutions to the existing problems. In turn, a certain discrepancy can be observed in area No. 3, where the number of social assistance services exceeds the number of reported problems. There are two possible explanations for the above: the situation has improved, and the number of welfare benefits is likely to decrease in the following year, or the existing social problems were overestimated, which led to the payment of higher benefits. These observations indicate that area No. 3 should be monitored closely in the future. The status of districts No. 6 and 13 and the surrounding areas should also be controlled. Despite the presence of social factors contributing to the risk of poverty, the number of welfare beneficiaries was very low or low in these areas. These districts should be analyzed in detail and included in prevention programs. Special measures should be also initiated to protect children in areas that are at risk of poverty. Districts marked in darker shades of blue are affected by social problems such as unemployment, lack of parenting skills, and alcoholism. The spatial distribution of different risk factors can be visualized on maps in successive analyses. Areas where the risk of social problems is particularly high due to neighborhood effects were also identified. For example, in areas No. 1 and 2, the accumulation of negative social phenomena is moderate, but the fact that these districts are neighbors can contribute to synergistic effects (negative phenomena will be mutually reinforced). In other cases, an area with a lower prevalence of social problems can be potentially threatened by the accumulation of negative phenomena in the immediate neighborhood. This situation can be observed in district No. 11 which remains under the direct influence of districts No. 4, 16, and 17. It should also be noted that neighborhood exerts a particularly strong effect on children—exposure to negative behaviors and social patterns during childhood affects life choices in adulthood [46,47,48]. For this reason alone, individuals (in particular families with children) residing in areas that remain under the negative influence of neighboring districts should be addressed by prevention programs.

The spatial distribution of the third dataset was visualized in Map 3 (Figure 5). This dataset contains a single component, namely families that require assistance with childcare and household tasks.

Figure 5 was developed in the GIS environment to illustrate the distribution of feature 18 (Table 1) determined by the Kernel Density method. The resulting map visualizes Personal assistance (SP 3, Table 1). Map 3 presents a special category of welfare services, where social workers regularly visit households and work with entire families. Assistance is provided to parents who lack parenting skills and are unable to meet their children’s needs. Some families also have poor household management skills; they have the necessary funds to cover basic life needs but are unable to manage household finances in a rational manner. This type of social assistance is rarely requested by beneficiaries, and the relevant needs are diagnosed by social workers during environmental assessments. Families with poor parenting and homemaking skills can receive support from a family assistant. Family assistants regularly monitor the family’s progress, help parents with planning and shopping for food and other products, give advice on childcare, or even teach family members to prepare healthy meals (Information about the services provided by family assistants was obtained during direct interviews with SAC employees in Bisztynek and Pieniężno). One family assistant supervises several to more than ten families; therefore, regular visits to the monitored households must be scrupulously planned.

The risk of domestic violence (dataset 4) is presented in Map 4 (Figure 6.). The severity of this social problem was determined based on the number of abuse victims receiving support, the number of police interventions responding to domestic violence, and the number of families covered by the Blue Card procedure. Social workers and the police intervene upon the request of domestic violence victims or based on witness reports. Various types of domestic violence are reported, and psychological abuse is often accompanied by physical violence, economic violence, and neglect. Families affected by domestic violence are officially registered in the Blue Card procedure. Such families should be regularly visited and monitored, and domestic abuse victims should receive support.

Figure 6 was developed in the GIS environment using Map Algebra with the Raster Calculator tool by overlaying the distribution of features 23, 24, and 25 (Table 1) determined by the Kernel Density method. The resulting map visualizes Violence (SP 4, Table 1). The employees of SACs have to maintain direct contact with welfare beneficiaries, including in their place of residence. Social workers make regular and planned visits to the supervised households, but also participate in interventions. Social assistance centers tend to be understaffed; therefore, the maps generated with the use of the proposed methods can be used by social workers to effectively plan their schedules and visit several households during a single trip to a given location.

The fifth dataset has prognostic value, and it is visualized in Map 5 (Figure 7). This dataset was used to identify welfare districts with a large senior population (The term “senior” denotes individuals who have reached the official retirement age which is set at 60 years for women and 65 years for men in Poland). An increase in the age dependency ratio resulting from the falling share of young people in the local population is particularly worrying in welfare districts that are situated remotely from major transportation routes and in spatially isolated communities. Seniors, in particular older persons living in single-person households, are at high risk of social exclusion. Such persons can experience negative effects of loneliness that are further exacerbated by poor health and declining mobility. Chronic loneliness contributes to addiction and depression. These risks could explain the steady increase in suicide rates among seniors [49].

Figure 7 was developed in the GIS environment using Map Algebra with the Raster Calculator tool by overlaying the distribution of features 26 and 27 (Table 1) determined by the Kernel Density method. The resulting map visualizes Aging (SP 5, Table 1). The age structure of welfare districts in a municipality should be regularly monitored to counteract these risks. Senior activity and wellness programs can be initiated in areas with a high age dependency ratio. According to the World Health Organization, active aging leads to better health and greater psychological and physical well-being [50].

## 5. Discussion 

Childhood is a time that should be particularly protected. Children should grow and develop in a safe and healthy environment. Therefore, social workers should attempt to eliminate social problems in specific areas. This study investigated the applicability of modern GIS tools for supporting the management of social welfare programs. Their applicability was expanded to include another group of users—SAC employees. The proposed solution was tested in a case study of a Polish municipality, and GIS tools were used to visualize the spatial distribution of social problems and persons in crisis.

It is worth noting that social welfare centers rarely use modern methods of data analysis. In most cases, the results are expressed in percentages which are used to calculate the relevant indicators, such as the ratio of long-term welfare recipients to the total number of welfare recipients. Social welfare organizations are understaffed in all countries around the world, and welfare employees rarely have experience in using data processing systems because most of them have degrees in pedagogy, special education, political science, social policy, psychology, sociology, or family science. Therefore, the methods for facilitating social work must be both effective and easy to use.

The fact that many countries, including Poland, have not developed cohesive databases of social problems poses a significant limitation in research dedicated to social phenomena. Data from different, unrelated sources were acquired for the needs of this study. The relevant information is not exchanged between institutions that generate such data. The authors did not have access to variables describing the magnitude of the analyzed problems, such as alcoholism. The relevant databases contain only information about the number of registered individuals with an addiction problem. These limitations further accentuate the need for an integrated database of the social problems identified in this study.

## 6. Conclusions

The key result of the present study was the observation that the nonparametric density estimator function in Esri ArcGIS software can be applied to process information about welfare beneficiaries and generate maps containing useful information. Social assistance involves temporary relief for persons in need, but also preventive measures and programs that enable individuals, families and entire communities to fully tap into their potential. Welfare support is essential for the achievement of sustainable development goals.

The study confirmed that GIS tools are useful for developing an MCA procedure for social sciences. An algorithm for visualizing five social phenomena was developed. This approach does not involve a prescribed method for compiling suitability or constraint maps because the Reclassify tool supports data transformation on the same scale, but the analyst must apply the same classification method. The analyst is also tasked with identifying the most effective method of qualifying the examined features/criteria. Above all, the same method should be applied to all raster reclassified data. Raw data (features) should also be selected at the analyst’s discretion. This is a useful option when the analyst does not have access to a full set of data or chooses different criteria than those presented in this article. It enables the analyst to select or eliminate key data describing a given social phenomenon. Despite the fact that the proposed tool is easy to use, the selection of features is a more difficult task; therefore, the analyst should have a working knowledge of the studied phenomena and the GIS environment. The tools available in the Arc GIS Pro program were used in this study. However, according to Abastante et al. [51], QGIS should be used for system deployments because this open-source system does not require a license, relies on readily available open data, and the georeferencing of objects can be assessed at any geographical scale (city, neighborhood, or single street), which ensures that the output is easy to read. The developed methodology can be used to analyze various social phenomena. The proposed algorithm is easy to implement, and it cuts the time and cost of analysis by comparing coefficients. To date, GIS tools have never been used to visualize social problems. Direct interviews with SAC employees revealed a general scarcity of tools for visualizing numerical data because only social indicators in databases are analyzed. The proposed procedure is innovative for several reasons. The social problem is defined, the associated features and criteria are determined, the identified features are subjected to a spatial analysis, and the results are visualized on a map. The values are transformed on a common scale, the layers are weighted and combined, a cartographic presentation is developed, and the results are analyzed. The usefulness of the proposed methodology was confirmed. The algorithm developed in the GIS environment facilitates semi-automatic generation of thematic maps of social problems on a single scale and in line with the adopted scheme, and it reduces analytical costs. Due to the availability and uncertainty of information, as well as the ambiguity of human perception and cognition, most selection parameters cannot be given accurately. The Fuzzy overlay method solves the Subjective weight problem [52]. The fuzzy set provides a mathematical model to determine the membership degree of each element to the set so that the applicability evaluation data of various subjective standards and the weight of the standard are converted into numerical values by the language of the decision maker [53,54].

The empirical results of the study revealed welfare districts with a high prevalence of social problems as well as areas whose welfare status could deteriorate due to the synergistic effects of problematic districts in the immediate neighborhood. The generated maps can be compared to make interesting observations. For example, a map presenting the number of registered income support beneficiaries in a given area was compared with a map illustrating areas that are at risk of poverty due to the presence of social factors that contribute to poverty.

The results of this study can be useful for SAC employees who are responsible for planning and managing social assistance services. Social workers can rely on the generated maps to plan visits to the supported families and households, monitor their progress and plan preventive measures. The present findings also have important implications for children from dysfunctional families. By identifying areas with the highest concentration of social problems, SACs can implement preventive programs to ensure that negative social patterns learned by children in dysfunctional families are not carried into adulthood. These observations are of great practical importance not only for Polish SACs, but also for welfare organizations in other countries. The proposed method is universal, and it can be applied in all types of territorial units. The presented GIS tool is flexible, and it can be easily adapted to the needs of other countries by incorporating country-specific welfare data in the analysis. The study has considerable practical relevance because it offers an effective method for adapting the existing GIS tools to new uses.

## Figures and Tables

**Figure 1 ijerph-19-07128-f001:**
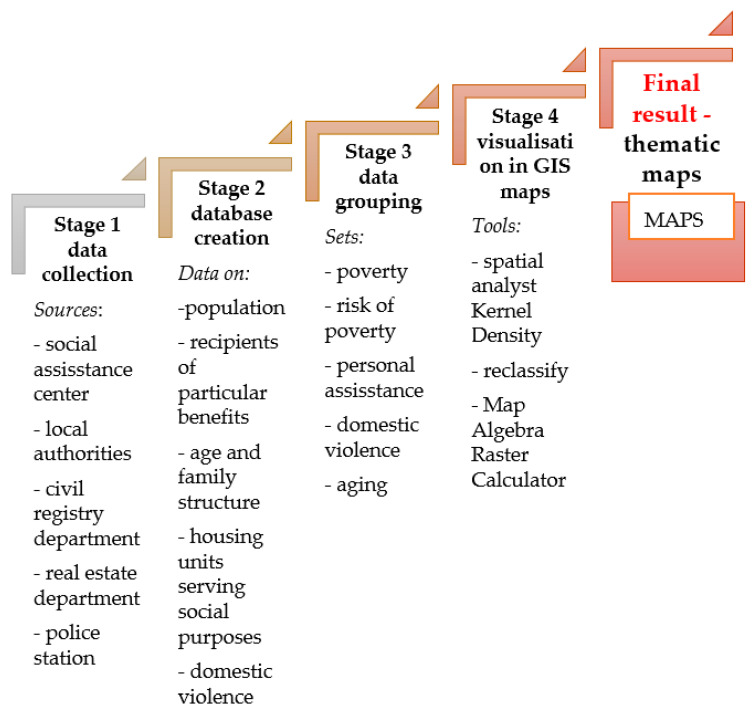
Research design algorithm. Source: Own elaboration.

**Figure 2 ijerph-19-07128-f002:**
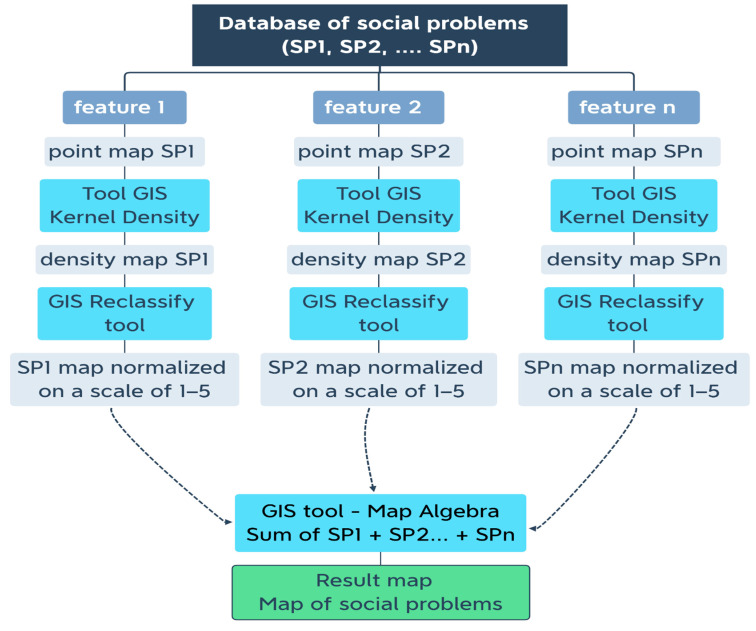
Algorithm developed in the GIS environment for mapping social problems. Source: Own elaboration.

**Figure 3 ijerph-19-07128-f003:**
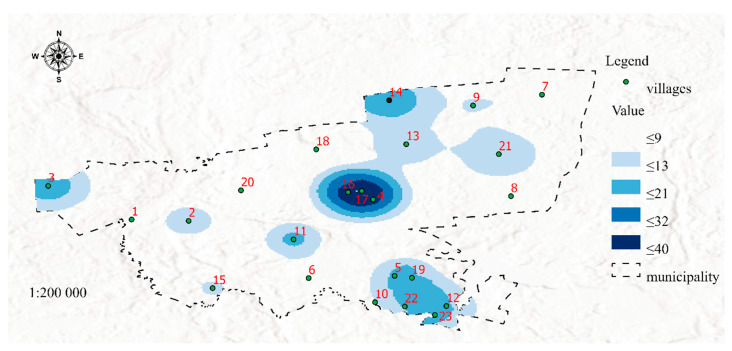
Map 1-Visualization of poverty areas (dataset 1). Source: Own elaboration.

**Figure 4 ijerph-19-07128-f004:**
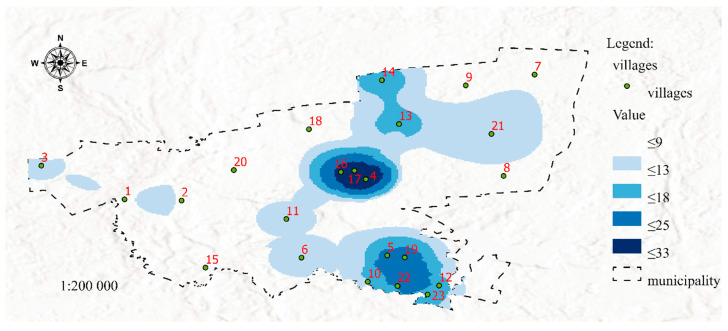
Map 2-Visualization of the future risk of poverty (dataset 2). Source: Own elaboration.

**Figure 5 ijerph-19-07128-f005:**
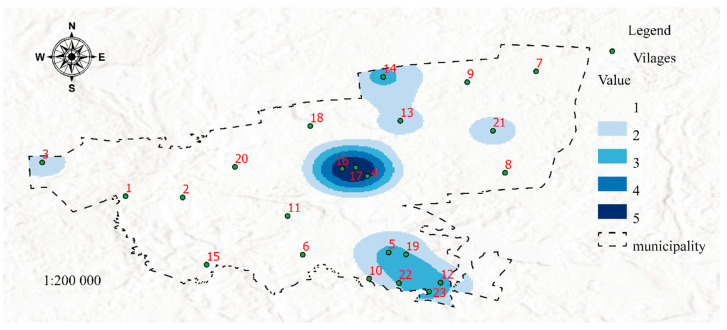
Map 3—Visualization of areas in need of personal assistance (dataset 3). Source: Own elaboration.

**Figure 6 ijerph-19-07128-f006:**
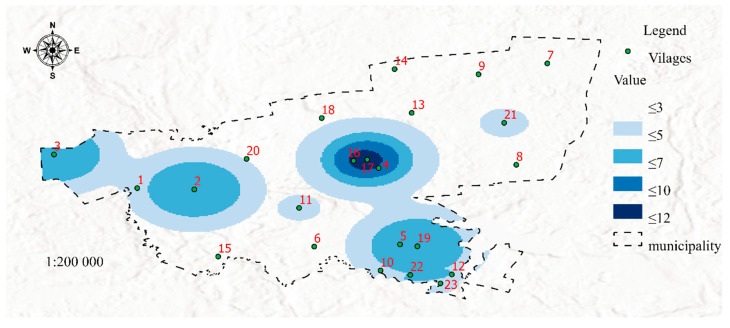
Map 4—Visualization of areas affected by violence (dataset 4). Source: Own elaboration.

**Figure 7 ijerph-19-07128-f007:**
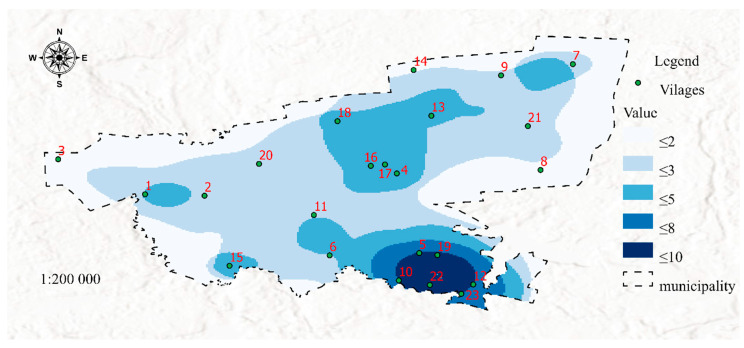
Map 5-Visualization of aging areas (dataset 5). Source: Own study.

**Table 1 ijerph-19-07128-t001:** Thematic groups of social problems. Source: Own elaboration.

Dataset	Social Problem	Social Assistance Targeting Specific Problems (Features)
**SP 1**	**Poverty**	- income support (F_1_) - permanent cash benefits (F_2_) - temporary cash benefits (F_3_) - special purpose allowances (F_4_) - noncash benefit-shelter (F_5_) - noncash benefit-food (F_6_) - noncash benefit-clothes (F_7_) - noncash benefit-care services (F_8_) - residence in a social welfare home (F_9_) - social housing (F_10_) - arrears in housing payments (F_11_) - orphanhood (F_12_) - evictions from social housing (F_13_) - homelessness (F_14_) - unemployment (F_15_) - disability (F_16_) - long-term or serious illness (F_17_) - lack of parenting and homemaking skills (F_18_) - alcoholism (F_19_) - drug addiction (F_20_) - difficulties with adjusting to life after release from prison (F_21_) - large families (F_22_)
**SP 2**	**Risk of poverty**	- orphanhood (F_12_) - evictions from social housing (F_13_) - homelessness (F_14_) - unemployment (F_15_) - disability (F_16_) - long-term or serious illness (F_17_) - lack of parenting and homemaking skills (F_18_) - alcoholism (F_19_) - drug addiction (F_20_) - difficulties with adjusting to life after release from prison (F_21_) - large families (F_22_)
**SP 3**	**Personal assistance**	- lack of parenting and homemaking skills (F_18_)
**SP 4**	**Violence**	- domestic violence (F_23_) - number of police interventions due to domestic violence (F_24_) - number of families covered by the Blue Card procedure (F_25_)
**SP 5**	**Aging**	- rate demographic of aging (F_26_) - aging ratio (F_27_)

## Data Availability

Not applicable.

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
