# Peer review of "Creating a Healthy Environment for Children: GIS Tools for Improving the Quality of the Social Welfare Management System"

_ijerph, 2022, doi:10.3390/ijerph19127128_

Round 1
Reviewer 1 Report
Although mentioned in the abstract, the introduction does not clarify how MCEs are considered in the research. In fact, the article proposes a multicriteria evaluation methodology but it is unclear what level of originality this method has and how it relates to GIS. In this regard, it is suggested that the following bibliography be observed:
- Abastante et al. 2020: Supporting resiliente urban planning through walkability assessment. Sustainability
- Lami et al (2014) Integrating multicriteria evaluation and data visualization as a problem structuring approach to support territorial transformation projects. EURO J. Decis. Process. 2014, 2, 281–312.
- Blecic et al (2015) Evaluating walkability: A capability-wise planning and design support system. Int. J. Geogr. Inf. Sci. 2015, 29, 1350–1374.
Line 239: An MCE algorithm is mentioned but is neither illustrated nor cited earlier in the text. It is not clear what the authors mean by MCE.
Figure 1 and 3 are not readable. It is suggested to increase the resolution (pixelated)
It is suggested that Table 1 be edited so that it is more readable
The formulas on lines 305, 306 and 309 (approximately) should be numbered. Also, being screenshots probably, they are not perfectly readable.
The conclusions should be more thorough and should demonstrate the usefulness of GIS and MCE (which is not considered in the conclusions) with respect to the subject matter. In this version the authors assert the usefulness of GIS but do not demonstrate it. Furthermore, although in the abstract the authors state the development of an MCE system by hinting at an innovative approach, this is almost completely missing throughout the article and conclusions.
Reviewer 2 Report
See my comments in the attached.

Round 2
Reviewer 1 Report
Dear authors,
thank you for improving the paper.
Regards